# Oncolytic Viruses and Immune Checkpoint Inhibitors: The “Hot” New Power Couple

**DOI:** 10.3390/cancers15164178

**Published:** 2023-08-19

**Authors:** Charlotte Lovatt, Alan L. Parker

**Affiliations:** 1Division of Cancer and Genetics, School of Medicine, Cardiff University, Heath Park, Cardiff CF14 4XN, UK; lovattce@cardiff.ac.uk; 2Systems Immunity University Research Institute, School of Medicine, Cardiff University, Heath Park, Cardiff CF14 4XN, UK

**Keywords:** oncolytic virus, immune checkpoint inhibitors, immunotherapy

## Abstract

**Simple Summary:**

Oncolytic viruses (OV) are engineered viruses designed to replicate selectively within tumour cells. They hold great promise as novel cancer therapeutics, but their performance clinically has, to date, failed to match expectations. One area of increasing interest in OV is the ability of these agents to induce lytic (“bursting”) cell death of tumour cells through replication. This lytic form of cell death is highly immunogenic, and therefore has the capacity to immunologically “heat up” otherwise “cold” tumours. Immune checkpoint inhibitors have revolutionized cancer immunotherapies, but frustratingly are only effective in a subset of patients with high levels of tumour infiltrating lymphocytes. There is increasing excitement that the combinations of OV with immune checkpoint inhibitors, or even immune checkpoint inhibitors encoded by OV, may prove synergistic, and have the potential to treat recalcitrant, immunologically cold tumours. Here, we review the evidence to date that such combination strategies may prove efficacious.

**Abstract:**

Immune checkpoint inhibitors (ICIs) have revolutionized cancer care and shown remarkable efficacy clinically. This efficacy is, however, limited to subsets of patients with significant infiltration of lymphocytes into the tumour microenvironment. To extend their efficacy to patients who fail to respond or achieve durable responses, it is now becoming evident that complex combinations of immunomodulatory agents may be required to extend efficacy to patients with immunologically “cold” tumours. Oncolytic viruses (OVs) have the capacity to selectively replicate within and kill tumour cells, resulting in the induction of immunogenic cell death and the augmentation of anti-tumour immunity, and have emerged as a promising modality for combination therapy to overcome the limitations seen with ICIs. Pre-clinical and clinical data have demonstrated that OVs can increase immune cell infiltration into the tumour and induce anti-tumour immunity, thus changing a “cold” tumour microenvironment that is commonly associated with poor response to ICIs, to a “hot” microenvironment which can render patients more susceptible to ICIs. Here, we review the major viral vector platforms used in OV clinical trials, their success when used as a monotherapy and when combined with adjuvant ICIs, as well as pre-clinical studies looking at the effectiveness of encoding OVs to deliver ICIs locally to the tumour microenvironment through transgene expression.

## 1. Introduction

The development of immune checkpoint inhibitor (ICI) therapies, which target immunosuppressive signals and restore anti-tumour immunity, has revolutionised the immunotherapy field in recent years. Antibodies targeting cytotoxic T lymphocyte-associated protein 4 (CTLA-4), programmed cell death protein 1 (PD-1) and its ligand PD-L1 aim to disrupt these negative regulatory signals, which under physiological conditions protect the host from autoimmunity and chronic inflammation, to disrupt the ability of tumour cells to evade the host immune response. The goal of ICI therapy is to recruit and activate innate and adaptive immune cells within the tumour microenvironment (TME), reverse T cell exhaustion, and reinvigorate anti-tumour T cells to control tumour growth [1]. As ICI therapy primarily functions to reinvigorate existing tumour reactive T cells, rather than induce their formation, durable clinical responses are most commonly seen in cancers which demonstrate an immunologically inflamed “hot” TME, characterised by a high somatic tumour mutation burden (TMB) and highly infiltrated immune active TMEs [2]. However, a lack of therapeutic benefit has been observed in those tumours which possess an immunologically “cold” TME; these tumours can either be immune desert TMEs, which demonstrate a low density of tumour infiltrating lymphocytes (TILs), or immune excluded TMEs, wherein T cells are localized at the invasive margins due to abnormal angiogenesis and an immunosuppressive stroma that prevents immune cell infiltration. In addition to poor T cell infiltration, “cold” TMEs are also characterised by low tumour mutational burdens, infiltration of immunosuppressive immune cells such as neutrophils, macrophages, myeloid-derived suppressor cells (MDSCs) and regulatory T cells (Tregs) and decreased antigen presentation and/or loss of tumour antigen presentation machinery [3,4].

Therefore, to effectively build on the recent successes of ICIs it is critical that, to extend their efficacy to non-responders, combination strategies need to be generated which aim to “heat up” tumours to obtain durable clinical responses. To this end, oncolytic viruses (OVs), which preferentially infect and destroy cancer cells, thus inducing immunogenic cell death, are a compelling combination agent which possess the ability to increase immune cell infiltration and overcome immunosuppression within the TME [5]. The promise of this combinatorial approach has led to multiple clinical trials which aim to investigate the efficacy of adjuvant OV and ICI therapy in several cancers. In this review, we describe the clinical efficacy of OVs as monotherapies and when delivered as a neoadjuvant with systemic ICI therapy. Furthermore, we explore the pre-clinical studies of OVs engineered to encode antibodies against immune checkpoints which aim to locally target ICI expression within the tumour to overcome the adverse events associated with systemic immunotherapy.

## 2. Oncolytic Viruses

OVs are immunotherapies which exploit the ability of replication-competent viruses to infect and replicate in tumour cells, whilst leaving healthy cells intact, leading to tumour cell lysis and subsequent release of viral progeny. Upon infection of cells, viruses possess the ability to promote their replication and subsequent release of viral progeny by interacting with cellular proteins to avoid immune cell recognition and early host cell death. Viruses typically activate one or more cell death pathways during infection and replication. Some forms of cell death are intrinsically tolerogenic and result in the uptake of dead cells by phagocytic cells; conversely, cell death can induce an innate and adaptive immune response termed immunogenic cell death (ICD). The induction of immunogenic tumour cell death results in local inflammation through the release of danger-associated molecular patterns (DAMPs) from dying infected cells, such as high mobility group box 1 (HMGB1), heat shock proteins (HSP), cell surface exposure of Calreticulin, and extracellular adenosine triphosphatase (ATP). Furthermore, virus replication and cell lysis leads to the release of pathogen-associated molecular patterns (PAMPs), such as viral proteins and nucleic acids, which further contribute to intensifying the immune response [6] (Figure 1).

This therapeutic efficacy is dependent on a fine balance between viral immunogenicity and anti-tumour immunity in which OVs can persist and avoid immune clearance, at least temporarily, to allow sufficient time for OVs to infect and replicate within tumour cells and to initiate an anti-tumour immune response [7]. In addition to their immunogenicity, the size, pathogenicity, and transgene capacity of a virus all contribute towards the selection of the appropriate vector for use as an OV therapy (Table 1). Some OVs, such as those derived from strains of coxsackie virus, influenza A virus (IAV), Newcastle disease virus (NDV), measles virus (MV), reovirus, vaccinia virus (VV) and vesicular stomatitis virus (VSV), demonstrate a natural tropism for tumours through exploitation of extracellular makers or dysregulated oncogenic intracellular pathways in tumour cells [8,9,10,11,12,13]. Furthermore, tumour cells often have defects in anti-viral mechanisms such as the type 1 interferon (IFN) pathway, thus further providing OVs such as NDV, VV and VSV with a replicative advantage [14]. Alternatively, OVs such as those derived from adenovirus (Ad) and herpes simplex virus (HSV) can be genetically modified to increase tumour cell selectivity through deletion and modification of genes to alter the natural tropism of the virus and provide a replicative advantage in tumour cells [14,15]. In addition, OVs can be further engineered through the insertion of eukaryotic transgenes to promote replication competence, limit their pathogenicity, increase their immunogenicity, and deliver additional genetic “payloads” which can promote anti-tumour immunity or increase the extent of tumour cell death [16].

### Turning Cold Tumours Hot: The OV Immune Response

During tumour development, tumour cells undergoing continuous remodelling at the genetic, epigenetic, and metabolic levels generate the critical modifications necessary for these cells to escape both innate and adaptive immune control, thus leading to malignant progression and growth of the tumour in the face of a competent immune system. Tumour immune evasion can result from changes at the level of the tumour, through inhibition of immune cell recognition and the selection of tumour variants that are resistant to immune effectors, or through the induction and recruitment of distinctive immunosuppressive immune cells and cytokines within the TME, thus generating a “cold” immunosuppressive TME [17]. The aim of immunotherapies is to increase T cell infiltration and revert these “cold” TMEs into immune activated and infiltrated “hot” TMEs indicative of an active anti-tumour immune response taking place; therefore, OVs which induce immunogenic tumour cell death and induce innate and adaptive immune responses are an ideal therapeutic candidate [18].

Following OV infection of tumour cells and subsequent local inflammation, innate immune cells, such as dendritic cells (DCs), natural killer (NK) cells and macrophages within the TME, recognize the DAMPs, PAMPs and tumour antigens released by oncolysis, resulting in the secretion of inflammatory cytokines, such as interferon-γ (IFN-γ), IFN-α, interleukin-6 (IL-6), IL-12 and tumour necrosis factor-α (TNF-α), which promote the maturation of DCs and further recruitment and activation of innate immune cells [19,20] (Figure 2). Antigen-loaded antigen-presenting cells (APCs) then migrate to draining lymph nodes where they initiate antigen-specific T cell priming and activation. In addition to T cell priming and activation, OV infection also elicits a potent type I IFN response, which stimulates the production of T cell-recruiting chemokines, which increase TME T cell infiltration [21]. Furthermore, the induction of inflammatory cytokines, such as TNF-α and IL-1β, upregulates the expression of selectin on endothelial cells, allowing for enhanced extravasation of T cells into the tumour [18]. Upon entering the TME, TILs must contend with an often dense network of stromal cells and extracellular matrix (ECM) which can prevent efficient T cell infiltration into the tumour. OV infection has been shown to alleviate these structural barriers through the recruitment of neutrophils which can secrete proteases, such as elastase and matrix metalloproteinases (MMPs), to degrade the ECM and increase immune cell infiltration [22,23]. In addition to the activation, priming, trafficking and infiltration of anti-tumour immune cells, OV infection can also overcome immunosuppressive signals within the TME through the stimulation of pro-inflammatory cytokine production and induction of potent pro-inflammatory M1 macrophages and type 1 helper (Th1) immune cell phenotypes [6,24,25].

## 3. Oncolytic Virus Monotherapy

To date, Ad, coxsackie virus, HSV, NDV, MV, VV and VSV OVs have all entered clinical trials for the treatment of several different cancers, with one adenovirus OV (H101) approved in China for the treatment of head and neck cancer, and two HSV OVs now approved for the treatment of metastatic melanoma in Europe and the US (T-VEC; Imlygic) and glioblastoma (GBM) in Japan (G47Δ; approval is conditional and time-limited based on verification and description of clinical benefit and safety in a post-market clinical study) (Table 2) [26,27]. Patients treated with OV monotherapy often demonstrate significant reductions in tumour burden, as demonstrated by decreased tumour size, partial response (PR), and complete response (CR) rates [9,28,29,30,31,32,33,34,35,36,37,38,39,40,41], or present with stable disease (SD) and progression free survival (PFS), suggesting that an anti-tumour immune response is occurring to control and destroy the tumour [8,9,27,28,29,30,32,36,37,38,42,43,44,45,46,47]. Indeed, OV treated tumours have demonstrated an increase in tumour infiltrating CD8+ T cells and the systemic presence of tumour antigen specific CD8+ T cells, in addition to a decrease in immunosuppressive MDSCs and Tregs within the TME [27,28,29,31,33,39,40,42,44,45,46,48]. However, despite demonstrating disease control and the presence of an activated immune response, only a small proportion of these clinical trials are able to demonstrate a durable clinical response (≥6 months) in small subsets of patients [27,28,29,31,35,44,45,47,49]. Furthermore, the failure of JX-594 to improve the survival or disease control rate (DCR) of hepatocellular carcinoma (HCC) patients who had failed first-line therapy with the multi-kinase inhibitor Sorafenib suggests that OVs may be more beneficial to a more fit patient population compared to those who are treatment-refractory [50].

Despite some promising examples of clinical efficacy, it is evident that OV monotherapies need to be enhanced for patient benefit. Several potential mechanisms, including the existence of neutralising antibodies, rapid anti-viral immune responses resulting in rapid and premature OV clearance, physical exclusion from the TME, and an immunosuppressive TME, may contribute to the modest activity seen with OV monotherapy and contribute to OV resistance [3]. Therefore, to enhance clinical efficacy, combined immunotherapeutic approaches, comprising OV with adjuvant ICI therapy, have been developed and demonstrate improved clinical efficacy when compared to either therapy alone [51].

## 4. Combined OV and ICI Therapy

### 4.1. Neoadjuvant Therapies

Expression of immune checkpoint molecules by cancer cells is one of the major mechanisms by which tumours can induce immunosuppression and subsequent immune evasion. CTLA-4, lymphocyte-activation gene 3 (LAG-3), T cell immunoglobulin and mucin-domain containing-3 (TIM-3), and T cell immunoreceptor with Ig and ITIM domains (TIGIT) all primarily interact with their ligands during the T cell priming stage, thereby limiting T cell activation, whilst PD-1:PD-L1 interactions occur predominantly in the periphery to regulate activated T cells during the effector phase [52]. CTLA-4 was the first immune checkpoint to be clinically targeted; it is expressed exclusively on Tregs and activated effector T cells, where it regulates the amplitude of T cell activation during priming; CTLA-4 expressing T cells often display tolerance towards tumours and CTLA-4 expressing Tregs contribute towards immunosuppression within the TME by further inhibiting the functions of other immune cells. Similar to CTLA-4 signalling, PD-1 binding with its ligands inhibits T cell function by reducing the intensity of IFN-γ, TNF and IL-2 production, reducing T cell survival through the inhibition of anti-apoptotic gene production, and suppressing T cell proliferation [53,54]. Such immune checkpoint molecule-mediated immunosuppression of the anti-tumour immune response facilitates the progression of cancer in the face of a competent immune system. Thus, ICI therapies aim to interrupt these immunosuppressive signals to restore anti-tumour immunity by exposing the tumour cells to a newly reinvigorated host immune response.

ICI antibodies such as the anti-CTLA-4 Ipilimumab, anti-PD-1 Nivolumab and Pembrolizumab, and anti-PD-L1 Atezolizumab have been approved for the treatment of several solid and haematological malignancies and have shown durable clinical responses in a proportion of patients [5]. In those patients who do benefit, clinical responses correlate with high tumour mutational burden, a rich neoantigen repertoire and the presence of a pre-existing anti-tumour response, as evidenced by increased TILs [7]. Conversely, those patients with cold TMEs, characterised by low tumour mutational burdens, a lack of expression or presentation of neoantigens, and low infiltration of TILs do not demonstrate durable clinical responses following treatment. As ICI therapy functions to reactivate an exhausted and suppressed anti-tumour immune response, TILs are the most important component for a patient to derive durable responses from ICI therapy, with the presence of a prominent T cell infiltration prior to treatment associated with increased sensitivity and survival following ICI treatment [55,56]. While several combination therapy approaches are in development to reverse these deficiencies in non-responsive patients, OV therapy is a promising combination therapeutic as it induces tumour cell death in a highly immunogenic context, thereby triggering an “in situ” tumour vaccination through the release of tumour antigens in the presence of virus-induced inflammation. Combination therapies using ICI and OVs are therefore attractive and potentially synergistic, as the OV therapy can “heat up” the TME by recruiting TILs, promoting further immune cell activation, and triggering the release of tumour antigens [57]. Moreover, treatment with Ad and HSV OV monotherapies have demonstrated significant increases in tumour PD-L1 expression, thus sensitizing tumours to subsequent ICI therapy [33,39,46].

Patients treated with neoadjuvant oncolytic Ad, HSV and VV followed by ICIs targeting PD-1, PD-L1 and CTLA-4 have all demonstrated clinical benefit, with durable response rates observed in subsets of patients (Table 3) [58,59,60]. Local intra-tumoral injection of OVs prior to systemic ICI therapy resulted in both an increase in CD8+ and CD4+ TILs and increases in circulating CD8+ and CD4+ T cells, in addition to local inflammation and reductions in the size of non-injected tumours, suggesting the presence of a systemic anti-tumour immune response [29,52,53,54,55,56,57,58,59]. When compared to Pembrolizumab monotherapy in advanced stage immunotherapy-naïve melanoma patients, combination T-VEC and Pembrolizumab therapy demonstrated slightly increased response rates and PFS, although this did not reach significance [60]. Similarly, combination T-VEC and Ipilimumab therapy did not significantly increase PFS or OS when compared to Ipilimumab alone; however, the combination therapy did demonstrate a significant increase in ORR (CR/PR). Furthermore, combination therapy resulted in an increase in the reduction in size of visceral non-injected lesions, consistent with a systemic anti-tumour immune response [61].

#### 4.1.1. Markers of Response

As expected, greater persistence of viral DNA in the tumour is indicative of greater clinical responses, with the presence of ONCOS-102 DNA at week 9 post-injection detectable in responders but undetectable in patients with progressive disease, suggesting that rapid OV clearance or less effective viral replication may prevent disease control [62]. In the same trial, the baseline presence of CD8+ and CD4+ TILs were also significantly greater in patients with disease control compared to those with progressive disease, with further increased tumour infiltration of CD4+ and CD8+ T cells following OV administration, only seen in patients with disease control [56]. Pre-treatment presence of CD3+/CD8+ aggregates at the infiltrating tumour edge and a greater abundance of TILs were also indicators of responsive patients in patients treated with T-VEC and Pembrolizumab [63]. However, objective responses to DNX-2401 and Pembrolizumab were only observed in patients with moderately inflamed TMEs, with those presenting pre-treatment with highly inflamed tumours enriched with exhausted immune cells, characterised by high expression of immunosuppressive immune checkpoints, receiving no improvements in survival, suggesting that the immunosuppressive TME in these patients may suppress any immune response induced by OV or ICI therapy [59]. The correlation with pre-treatment immune infiltration and response to combination therapy raises some concerns, as although the neoadjuvant OV therapy aims to induce immunogenic cell death and increase immune infiltration into the tumour, it appears that the tumour must already have some evidence of an immune response in order to demonstrate a response. This suggests that, as with ICI therapy, those patients who present with immunologically cold tumours may not derive good clinical responses from these therapies.

**Table 3 cancers-15-04178-t003:** Neoadjuvant OV and ICI therapy trials and their key findings.

Virus	OV	ICI	Indication	Key Findings	Ref.
Ad	CG0070 (IVS)	PD-1: Pembrolizumab	NMIBC Phase II	*Disease control:* 82% 6-month CR; 81% 9-month CR; 68% 12-month CR	
	DNX-2401 (IT)		GBM Phase II	*Survival:* 52.7% 12-month survival; 12.5 months median OS; 3 patients alive > 45 months *Disease control:* ORR 10.4%; 42.9% SD; 4.8% CR; 7.1% PR;	[59]
	EnAd (IT)	PD-1: Nivolumab	mCRC Phase I	*Survival:* median OS 15.4 months (5 months placebo) *Disease control:* median PFS 2.8 months *Immune response:* 85% demonstrated increased CD8+ TILs; 77% increased CD4+ TILs; 62% increased PD-L1+ TILs	[64,65,66]
	ONCOS-102 (IT)	PD-1: Pembrolizumab	Melanoma progressing post-PD-1 blockade Pilot	*Disease control:* 35% ORR; 64% SD; 27% demonstrated CR in injected tumour 53% demonstrated reduction in ≥1 non-injected tumour *Immune response:* increased CD4+ & CD8+ TILs	[62]
HSV	T-VEC (IT)	PD-1: Pembrolizumab	Melanoma Phase Ib	*Disease control:* 82% demonstrated >50% reduction of injected tumours; 43% in non-injected tumours *Immune response:* 67% demonstrated increased CD8+ TILs; demonstrated increased systemic proliferating CD8+ T cells	[67]
			Melanoma Phase III T-VEC + Pemb vs Pemb	*Disease control:* T + P: 17.9% CR; 48.6% ORR (CR/PR); 14.3 months PFS P: 11.6% CR; 41.3% ORR; 8.5 months PFS	[60]
			Sarcoma Phase II	*Disease control:* 21% PR; 47% SD; median PFS 17.1 months *Immune response* responders saw increased CD8+ TILs and CD8+ aggregates at tumour edge; non-responders saw no increase in CD8+ TILs or aggregates	[63]
		CTLA-4: Ipilimumab	Melanoma Phase II TVEC + Ipi vs Ipi	*Disease control:* T + I: 13% CR; 26% PR; 39% ORR (CR/PR); 8.2 months median PFS; 52% non-injected visceral tumour reduction I: 7% CR; 11% PR; 18% ORR; 6.4 months median PFS; 23% non-injected visceral tumour reduction	[61]
	HF10 (IT)	CTLA-4: Ipilimumab	Melanoma Phase II	*Survival:* Median OS 26 months *Disease control:* median PFS 19 months; 68% SD *Immune response:* increased CD8+ and decreased CD4+ TILs	[34]
VV	JX-594 (IT)	CTLA-4: Tremelumab PD-L1: Durvalumab	ICI refractory CRC Phase I/II	*Survival:* J + D: Median OS 7.5 months J + D + T: Median OS 5.2 months *Disease control: J + D:* median PFS 2.3 months; 12.5% DCR J + D + T: median PFS 2.1 months; 16.7% DCR *Immune response:* Increased proliferating CD3+ TILs after OV treatment and again after ICI treatment; increased M1 macrophages in tumours	[68]

Abbreviations: Ad, adenovirus; CRC, colorectal cancer; DRC, disease control rate; GBM, glio-blastoma; HSV, herpes simplex virus; ICI, immune checkpoint inhibitor; IT, intra-tumoural; IVS, intra-vesicular; mCRC, metastatic colorectal cancer; NMIBC non-muscle-invasive bladder cancer; ORR, overall response rate; OS, overall survival; OV, oncolytic virus; PD-1, programmed cell death protein 1; PD-L1, programmed death-ligand 1; PFS, progression free survival; PR, partial response; SD, stable disease; TIL, tumour infiltrating lymphocyte; VV, vaccinia virus.

#### 4.1.2. Adverse Events

Although all studies reported adverse effects in both the monotherapy and combination therapy arms, none reported dose-limiting toxicities, and all adverse events (AEs) were those expected and observed with the single-agent use of either therapy. Combination treatments were not associated with an increase in the incidence or severity of AEs and most AEs, such as fatigue, fever, chills, arthralgia, rash and nausea, were of mild to moderate (Grade 1/2) severity. Grade 3/4 AEs occurred in subsets of patients, in addition to some fatal AEs; however, when compared to monotherapies, again the incidence of these events was similar, suggesting that combination therapy is tolerable for patients [29,52,53,54,55,56,57,58,59,60]. ICI therapy functions to remove the inhibitory signals placed on effector immune cells, thus effectively removing the brakes on the anti-tumour immune response; however, although OVs are often locally delivered by intra-tumoural injection, ICIs are delivered systemically via the intravenous route. Under normal physiological conditions, immune checkpoints function to prevent over activation of the immune system and auto-immune responses, therefore systemic blocking of these signals often results in autoimmune AEs. These systemic toxicities associated with intravenous ICI therapy could potentially be reduced by targeted delivery of the antibodies directly into the TME; indeed, low-dose intra-tumoural administration of ICIs has been shown to be comparable to systemic high-dose delivery [69,70]. Therefore, the use of OVs engineered to express ICIs, thus limiting ICI expression to areas of viral replication within the TME, is an attractive approach for local ICI delivery, which may limit systemic AEs.

### 4.2. OVs Encoding ICIs

Ad, HSV, IAV, NDV, MV, VV, VSV and chimeric poxviruses have all been engineered to express ICIs targeting CTLA-4, PD-1 and PD-L1, either as full-length IgG antibodies, single-chain fragment variables (scFV), or scFV-Fc fusion proteins, showing promising results in a variety of in vivo animal models (Table 4). In comparison to full length monoclonal antibodies (mAbs), scFvs are fragments of antibody consisting of variable regions of the light (VL) and heavy (VH) chains joined by a flexible linker peptide; this smaller size allows for greater penetration and efficient localisation into the tumour, and faster clearance from the blood. Furthermore, the reduced transgene size can allow for the addition of further transgenes in OVs with high loading capacity, thus further enhancing the immunotherapeutic viral payload [57]. Despite antibodies naturally being produced in highly specialized and differentiated plasma cells, it has been demonstrated that functional ICI antibodies can be detected in tumour cells in vitro [71,72,73,74,75,76,77,78] and in vivo [72] following infection with engineered OVs, and that these antibodies are therapeutically functional.

Intra-tumoural injection of ICI-encoding OVs (ICI-OVs) resulted in significant reductions in tumour volume compared to untreated [46,71,72,79,80,81,82,83] and parental OV treated tumours [74,75,76,77,78,84,85,86,87,88], with ICI-OVs also significantly increasing OS compared to untreated [80,81,82,83,89,90] and parental OV treated tumours [75,77,79,84,86,87,88,91], demonstrating that OVs encoding for antibodies against CLTA-4, PD-1 and PD-L1 significantly increase the anti-tumour immune response. This increase in therapeutic activity was associated with increased CD4+ and CD8+ T cell and decreased Treg tumour infiltration, and an increase in the proportion of activated and effector memory and central memory T cells [76,77,78,86,88,91,92]. Furthermore, local intra-tumoural administration of ICI-OVs elicited pronounced systemic increases in activated CD4+ and CD8+ T cells, increased effector memory and central memory T cells and decreased MDSC and Treg populations [75,79,83]. Moreover, an abscopal effect mediated by a systemic anti-tumour response was observed using bilateral tumour models with unilateral IT injection, wherein un-injected distant tumours demonstrated decreased tumour growth and increased immune infiltration and activation [76,77,92], with one melanoma model demonstrating delayed non-injected tumour growth and prolonged survival with IAV-CTLA4 compared to parental IAV treatment [74]. The delay in untreated tumour growth, in combination with the increased presence of memory T cells, is indicative of the induction of long-term immune memory. Indeed, when re-challenged with tumour cells in vivo, ICI-OV treated animals were able to successfully inhibit tumour growth [75,76,79,83,84], with one study demonstrating significant reductions in tumour growth and increased survival following VV-PDL1 treatment compared to parental VV treatment [92]. Taken together, these pre-clinical studies demonstrate that OVs encoding ICI antibodies can significantly decrease tumour growth and prolong survival when compared to parental OV treatment in multiple tumour models.

When compared to parental OV plus systemic ICI treatment, ICI-OVs demonstrate similar reductions in tumour volume and increased survival, suggesting that ICI-OVs could represent a promising new immunotherapeutic with reduced AEs compared to OV + systemic ICI [75,78,79,84,91,92,93]. Currently, research into IVI-OVs is predominantly limited to mouse models and data on AEs is limited at present. However, the safety profile of IV injection of an oncolytic HSV1 encoding a PD-1 scFv was assessed in a more clinically relevant non-human primate model and demonstrated no abnormal body weight or temperature changes, slight elevations in serum markers of renal and liver dysfunction which returned to normal after several days, no overt changes in leukocyte counts or increases in cytokine production, and no obvious pathological abnormalities in any organs [77]. These favorable safety outcomes in a non-human primate model, in combination with the toxicity studies in humanized mouse models, are an encouraging step in the translation of these ICI-OVs to the clinic, wherein the full therapeutic benefit of a human OV encoding for a human ICI can be studied.

However, it should be noted that there could be some limitations to ICI-OVs, such as “on target, off tumour” activity. Although OVs are selected either for their natural tumour selectivity or through modification of the OVs tropism to target tumour-specific markers, there is evidence that OVs can infect healthy cells. However, modification of the OV genome to either increase tumour selectivity through the addition of tumour-selective promoters, or insertion of the ICI transgene into late transcription units, or via alternative splicing, to ensure replication-dependent ICI expression can reduce off target activity [57,94]. Furthermore, ICIs are currently given as a systemic therapy, therefore the effects of local production within subsets of “on target” healthy cells would be similar to that seen with the current ICI treatments. Additionally, a benefit of systemic ICI therapy is the ability to stop treatment if adverse events and toxicities are observed. OV replication efficiency is difficult to predict in individual patients, therefore the addition of safety switches, such as the tetracycline-derived “tet system” which leads to gene repression in the presence of tetracycline, within the OV genome, which either block OV replication or antibody expression upon the presentation of toxicities, will be an important area of research in the ICI-OV field [57,95].

**Table 4 cancers-15-04178-t004:** Pre-clinical trials of OVs engineered to express ICI antibodies and their key findings.

OV	Target	ICI Format	Indication	Key Findings	Ref.
Ad5	CTLA-4 mouse	IgG2	Melanoma NSCLC SCLC	**Subcutaneous mouse xenograft model with intravenous OV injection**: *Disease control:* significant 72% reduction in tumour growth compared to untreated tumours **Subcutaneous mouse xenograft model with intra-tumoural OV injection:** *Disease control:* significant 3-fold decrease in tumour growth compared to untreated tumours	[71]
Ad5/3	CTLA-4 human	IgG2	NSCLC Prostate	**Subcutaneous T-cell-deficient mouse xenograft model with intra-tumoural OV injection: ***Disease control:* significantly decreased tumour growth compared to untreated *Immune response:* 43-fold increase in tumour anti-CTLA-4 antibody concentrations compared to systemic plasma **In vitro human T cell activation assay***:* PBMCs from advanced solid cancer patients cultured in the presence of supernatant from OV-infected cells saw increase in T cell IL-2 and IFN-γ production	[72]
HSV-1	CTLA-4 & GM-CSF mouse	scFv fused to mouse IgG1	Lymphoma	**Bilateral subcutaneous mouse xenograft model with single-sided intra-tumoural OV injection:** *Disease control:* decreased tumour growth in both injected and non-injected tumours (not significant)	[73]
IAV	CTLA-4 mouse	scFV	Melanoma	**Bilateral subcutaneous mouse xenograft model with single-sided intra-tumoural OV injection:** *Disease control:* significantly decreased tumour growth in both injected and non-injected tumours and prolonged survival compared to parental virus	[74]
IAV	CTLA-4 mouse	scFV	HCC	**Spontaneous homograft model with intra-tumoural OV injection***:* *Survival:* prolonged survival compared to parental OV *Disease control:* significantly decreased tumour compared to parental OV	[85]
NDV	CTLA-4 mouse	scFV	Melanoma	**Intradermal mouse tumour model with intra-tumoural OV injection**: *Survival:* prolonged survival compared to systemic CTLA-4 treatment plus parental NDV *Disease control:* comparable tumour growth inhibition	[93]
MV	CTLA-4 mouse	scFV-IgG1 Fc fusion	Melanoma	**Subcutaneous synergic mouse tumour model with intra-tumoural OV injection**: *Disease control:* significantly decreased tumour growth compared to parental virus and untreated *Immune response:* significant increase in tumour T cell infiltration and a decrease in Treg infiltration compared to parental OV and untreated; increased splenocyte IFN-γ release upon re-stimulation with tumour cells in vitro compared to parental OV and untreated	[91]
Ad68	PD-1	IgG4	Colorectal	**Bilateral subcutaneous humanised PD-1 transgenic mouse tumour model with single-sided intra-tumoural OV injection***:* *Survival:* prolonged survival compared to parental OV and untreated *Disease control:* significantly decreased tumour growth compared to parental OV and untreated, with successful tumour rejection upon rechallenge *Immune response:* significantly increased systemic CD8+ T cell and effector and central memory T cell proportions; significantly decreased PD-1+ CD4+ and CD8+ T cell proportions	[75]
HSV-1	PD-1 mouse	scFv	HCC	**Bilateral subcutaneous synergic mouse tumour model with single-sided intra-tumoural OV injection:** *Disease control:* significantly decreased tumour growth in both injected and non-injected tumours and greater long-term tumour growth inhibition compared to parental OV and untreated; successful tumour rejection upon rechallenge *Immune response:* significantly increased activated CD4+ and CD8+ cell tumour infiltration compared to parental OV; however, also saw significantly greater MDSC infiltration compared to parental OV	[76]
HSV-1	PD-1 human	scFv	HCC	**Orthotopic HCC xenograft tumour model with intravenous OV injection in humanised PD-1 transgenic mice***:* *Survival:* and increased overall survival compared to parental OV and untreated mice *Disease control:* significantly decreased tumour growth compared to parental OV and untreated mice, with all anti-PD-1 OV treated mice tumour free at 12 weeks **Bilateral subcutaneous mouse xenograft tumour model with single-sided intra-tumoural OV injection in humanised PD-1 transgenic mice:** *Disease control:* significantly decreased tumour growth in both injected and non-injected tumours compared to parental OV and untreated *Immune response:* anti-PD-1 OV treated tumours demonstrated significantly reduced proportions of exhausted CD8+ T cell populations and increased effector memory CD8+ T cell populations compared to parental OV and untreated	[77]
HSV-1	PD-1 human	scFv	Melanoma	**Bilateral subcutaneous mouse xenograft tumour model with intra-tumoural OV injection in humanised PD-1 transgenic mice***:* *Disease control:* significantly decreased tumour growth compared to untreated and parental OV *Immune response:* significantly increased tumour CD4+ and CD8+ T cell infiltration compared to untreated; RNA-seq analysis demonstrated significant enrichment in anti-viral, IFN and antigen presentation and processing pathways compared to untreated	[78]
HSV-1	PD-1 human	scFV	GBM	**Orthoptic GBM synergic mouse tumour model with intra-tumoural OV injection***:* *Survival:* increased median survival time compared to untreated (significant) and parental OV (not significant) *Disease control:* successful tumour rejection following rechallenge	[89]
HSV-2	PD-1 human	IgG	Melanoma	**Subcutaneous mouse xenograft tumour model with intra-tumoural OV injection in humanised PD-1 transgenic mice:** *Survival:* prolonged survival compared to untreated; improved tumour-free survival compared to parental OV and untreated *Disease control:* significantly decreased tumour growth compared to untreated; successful tumour rejection following rechallenge *Immune response:* increased systemic percentages of CD4+, CD8+ and CD3+ T cells and significant increase in T cell activation markers compared to parental OV and untreated; significant reduction in Tregs and MDSCs compared to untreated	[79]
VV	PD-1 mouse	IgG & scFV	Fibrosarcoma Melanoma	**Subcutaneous synergic mouse tumour model with intra-tumoural OV injection:** *Survival:* prolonged survival (IgG significant; scFV not significant) compared to parental OV and untreated *Disease control:* significantly decreased tumour growth compared to parental OV and untreated *Immune response:* IgG-OV significantly increased tumour infiltration of CD4+ and CD8+ T cells, the proportion of activated CD8+ T cells, and the CD8+/Foxp3+ T cell ratio compared to systemic anti-PD-L1 treatment, but to a lesser extent than parental OV alone	[86]
NDV	PD-1 and PD-L1 mouse & IL-2	scFV	Melanoma	**Unilateral subcutaneous synergic mouse tumour model with intra-tumoural OV injection: ***Survival:* prolonged survival compared to parental OV *Disease control:* significantly decreased tumour growth compared to parental OV **Bilateral subcutaneous synergic mouse tumour model with single-sided intra-tumoural OV injection:** *Survival:* when combined with systemic anti-CTLA-4 treatment, PD-1 and PD-L1 OV demonstrated significantly prolonged survival compared to parental OV *Disease control:* when combined with systemic anti-CTLA-4 treatment, PD-1 and PD-L1 OV demonstrated significantly inhibited tumour growth in non-injected tumours compared to parental OV	[87]
MV	PD-1 & PD-L1 mouse	scFV-IgG1 Fc fusion	Melanoma	**Subcutaneous synergic mouse tumour model with intra-tumoural OV injection**: *Survival:* significantly prolonged survival compared to parental OV and untreated *Disease control:* significantly decreased tumour growth compared to parental OV and untreated; successful tumour rejection following rechallenge *Immune response:* significantly increased activated CD8+ T cell and reduced Foxp3+ Treg tumour infiltration; higher effector memory T cell: central memory T cell ratio for PD-1 (significant) and PD-L1 (not significant) OVs compared to untreated	[84,91]
Ad5/24	PD-L1 mouse	scFV	Colorectal	**Bilateral subcutaneous synergic mouse tumour model with intra-tumoural OV injection**: *Survival:* significantly prolonged survival compared to parental OV and untreated *Disease control:* significantly decreased tumour growth compared to parental OV and untreated *Immune response:* significantly increased tumour CD8+ T cell infiltration compared to parental OV	[88]
Chimeric poxvirus	PD-L1 human	scFv	Breast cancer Gastric cancer PDAC	**Orthotopic synergic mouse breast cancer model with intra-tumoural or intravenous OV injection***:* *Survival:* significantly prolonged survival compared to untreated *Disease control:* significantly decreased tumour growth compared to untreated **Orthotopic mouse breast cancer xenograft model with intra-tumoural OV injection:** *Survival:* significantly prolonged survival compared to untreated *Disease control:* significantly decreased tumour growth compared to untreated **Peritoneal mouse GC and PDAC xenograft tumour model with intraperitoneal OV injection:** *Survival:* significantly prolonged survival compared to untreated *Disease control:* significantly decreased tumour growth compared to untreated	[80,81,82,90]
VSV	PD-L1 human	scFV	Lung carcinoma	**Subcutaneous mouse hPD-L1 knock-in synergic tumour model with intra-tumoural OV injection:** *Survival:* significantly prolonged survival compared to untreated *Disease control:* significantly decreased tumour growth compared to untreated successful tumour rejection following rechallenge *Immune response:* significant systemic increase in total number of CD8+ effector memory and CD8/CD4+ central memory T cells	[83]
VV	PD-L1 & GM-CSF human	Soluble PD-1 ED fused to IgG1 Fc	Melanoma	**Bilateral subcutaneous synergic mouse tumour models with intra-tumoural OV injection:** *Survival:* significantly decreased prolonged survival upon tumour rechallenge compared to untreated and parental OV *Disease control:* decreased tumour growth in 3 solid tumour models; significantly decreased tumour growth and prolonged survival upon tumour rechallenge compared to untreated and parental OV *Immune response:* significantly increased CD45+, DC, CD4+ and CD8+ T cell, and decreased MDSC and Treg tumour infiltration in injected tumours; untreated distant tumours also demonstrated increased infiltration and activation of lymphocytes and other immune cells	[92]

Abbreviations: Ad, adenovirus; CTLA-4, cytotoxic T-lymphocyte associated protein 4; DC, dendritic cell; GBM, glioblastoma; GM-CSF, granulocyte-macrophage colony-stimulating 2; HCC, hepatocellular carcinoma; HSV, herpes simplex virus; IAV, influenza A virus; IFN, interferon; IL-2, interleukin-2; MDSC, myeloid-derived suppressor cell; MV, measles virus; NDV, Newcastle disease virus; NSCLC, non-small cell lung carcinoma; OV, oncolytic virus; PBMC, peripheral blood mononuclear cell; PD-1, programmed cell death protein 1; PD-L1, programmed death-ligand 1; PDAC, pancreatic ductal adenocarcinoma; scFV, single chain variable fragment; SCLC, small cell lung carcinoma; treg, regulatory T cell; VSV, vesicular stomatitis virus; VV, vaccinia virus.

#### Additional Targets

Activation of an immune response following ICI therapy has been associated with the upregulation of additional immune checkpoint molecules, such as CTLA-4, PD-1, PD-L1, TIGIT, LAG-3 and TIM-3 on immune cells [96]. This increase in immunosuppressive signals is one such mechanism by which patients can become resistant to ICI therapy; therefore, combination therapies targeting multiple immune checkpoints are a promising mechanism to improve therapeutic outcomes, as by targeting multiple immunoregulatory pathways the likelihood of a successful and sustained anti-tumour immune response is increased [97]. Indeed, data from the ICI-OV pre-clinical studies has suggested that the therapeutic outcome of these ICI-OVs can be further improved with the addition of systemic ICIs targeting additional immune checkpoints. An oncolytic NDV encoding for an anti-PD-1 scFV demonstrated significant survival benefits over the parental virus when combined with systemic CTLA-4 treatment, with the combination therapy targeting two immunoregulatory pathways at distinct yet synergistic stages in the immune response, namely the priming (CTLA-4) and effector (PD-1) phases of the adaptive immune response, inducing up to 50% CR rates [87]. Likewise, systemic targeting of TIM-3, an immune checkpoint that functions to suppress T cell responses, has been shown to more potently suppress tumour growth and improve the anti-tumour efficacy of an anti-PD1 scFV HSV OV [77]. In a different study, the anti-tumour efficacy of an anti-PD-1 scFV HSV OV was improved by the addition of systemic TIGIT ICI therapy, as evidenced by increased splenic tumour-specific CD8 T cells [76]. TIGIT is expressed on naïve T cells and activated NK cells and Tregs, and interacts with its two major ligands, poliovirus receptor (PVR; CD155) and poliovirus receptor-related 2 (PVRL2; CD112), which are expressed on myeloid cells and tumour cells [98]. This enhanced therapeutic efficacy demonstrated with combinatorial PD-1 and CTLA-4/TIM-3/TIGIT blockade demonstrates that immune checkpoints which function in distinct yet synergistic stages in the immune response, namely the priming (CTLA-4/TIM-3/TIGIT) and effector (PD-1) phases of the adaptive immune response, can synergise to enhance anti-tumour immunity. In addition to systemic anti-TIGIT therapies, an oncolytic VV armed with an scFV against TIGIT has demonstrated enhanced anti-tumour efficacy and increased recruitment and activation of T cells within the TME compared to parental virus in several subcutaneous tumour models [99,100]. Research into additional novel immune checkpoints, such as LAG-3, CD200, TIM-3, and B7 homolog 3 protein (B7-H3), have shown promising results in pre-clinical and clinical models, suggesting that OVs could be engineered to express antibodies against these checkpoints in the future [101].

## 5. Conclusions

Monoclonal antibodies targeting the immune checkpoints CTLA-4, PD-1 and PD-L1 have demonstrated promising clinical efficacy in several cancers, with subsets of patients deriving durable clinical responses. However, as PD1/PD-L1 ICI therapy works to re-invigorate tumour reactive T cells, its success is based on the presence of a pre-existing anti-tumour immune response and an immunologically “hot” TME. Thus, overall response rates are 47–63%, with non-responder patients demonstrating both cell intrinsic and extrinsic primary resistance mechanisms. Similarly, response rates with CTLA-4 ICI therapy are 10–20% [102,103]. Furthermore, of those that do show an initial or sustained response, disease relapse and progression occur in most cases due to acquired secondary resistance mechanisms within the TME. Therefore, therapeutic combinations which can turn an immunologically “cold” tumour into a highly inflamed “hot” tumour which is primed for subsequent ICI therapy are a promising mechanism to overcome ICI resistance. OVs which selectively replicate within and kill tumour cells, resulting in the induction of immunogenic cell death and the augmentation of anti-tumour immunity, have emerged as a promising modality for combination therapy.

OV monotherapies have demonstrated moderate clinical efficacy in several clinical trials and, when used as a neoadjuvant with systemic ICIs, have been shown to increase the anti-tumour immune response. There are currently several ongoing clinical trials in a range of cancers looking at novel neoadjuvant OV and ICI combination therapies with the aim of achieving durable clinical responses in patients who would often not benefit from systemic ICI treatment alone. In addition, pre-clinical animal models of OVs engineered to encode for ICI antibodies have shown promising results, with tumour growth control and overall survival similar to that seen with OV and systemic ICI therapy. A clinical trial looking at the efficacy of an oncolytic adenoviral vector encoding an anti-CD40 antibody in advanced tumours, alone and in combination with systemic Pembrolizumab, has been completed and is currently awaiting results (NCT03852511). This targeted and local expression of ICI antibodies could represent a mechanism by which the AEs associated with systemic alleviation of immunosuppression, a major drawback to ICI therapy, can be overcome; therefore, these results are eagerly anticipated.

## Figures and Tables

**Figure 1 cancers-15-04178-f001:**
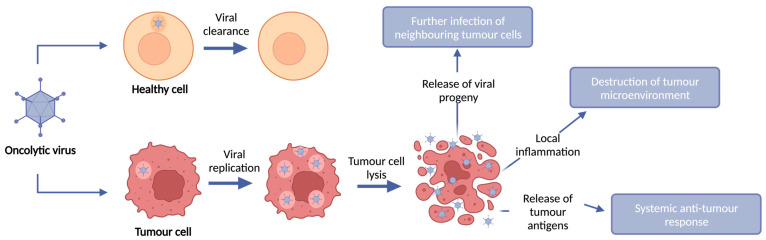
OV anti-tumour mechanism of action. OVs selectively infect, replicate within, and lyse tumour cells whilst leaving healthy cells intact. Upon infection of tumour cells, OVs replicate and lyse the tumour cell, resulting in tumour cell death and release of viral progeny. This tumour cell lysis results in destruction of the local tumour microenvironment and induction of an anti-tumour immune response through local immune infiltration and release of tumour antigens.

**Figure 2 cancers-15-04178-f002:**
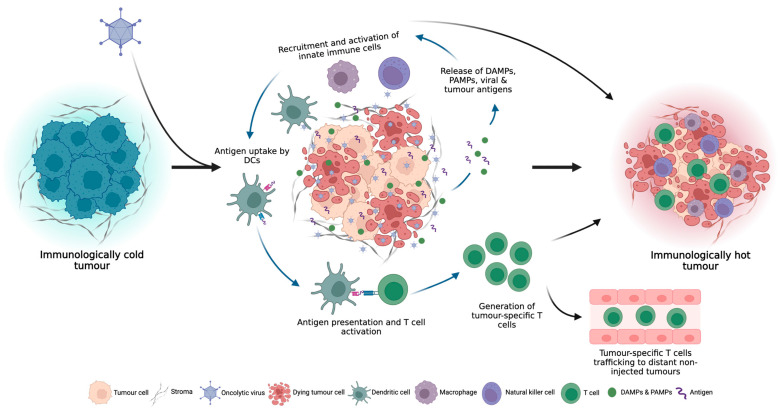
OV infection can turn immunologically “cold” tumours, which do not respond well to ICI therapy, to immunologically “hot” tumours through the induction of local and systemic anti-tumour immune responses.

**Table 1 cancers-15-04178-t001:** Viruses commonly used as OV vectors and their features.

Virus	Diameter	Genome	Genome Size	Transgene Capacity
Adenovirus	90–100 nm	dsDNA	30–36 kb	~2.5 kb
Herpes simplex virus	200 nm	dsDNA	~152 kb	~30 kb
Vaccinia virus	350 nm	dsDNA	~192 kb	~25 kb
Influenza A virus	80–120 nm	ss(–)RNA	~13.5 kb	~2.4 kb
Newcastle disease virus	100–500 nm	ss(–)RNA	~15 kb	~4.5 kb
Measles virus	100–200 nm	ss(–)RNA	~16 kb	~6 kb
Vesicular stomatitis virus	70–200 nm	ss(–)RNA	~11.1 kb	~4.5 kb
Coxsackie virus	22–30 nm	ss(+)RNA	~7.5 kb	<1 kb
Reovirus	80 nm	dsRNA	24 kb	~1.5 kb

ds, double stranded; ss, single stranded; kb, kilobase.

**Table 2 cancers-15-04178-t002:** OV clinical trials and their key findings.

Virus	OV	Engineered Specificity	Transgene	Indication	Delivery	Key Findings	Ref.
Adenovirus	CG0070	Ad5 with *E1a* under *E2F-1* promoter	GM-CSF	NMIBC Phase II	IVS	*Disease control:* 47% 6-month CR; 29% 12-month CR	[49]
	DNX-2401	Ad5 with 24 bp *E1a* deletion; RGD integrin-binding motif		GBM Phase I	IT	*Survival:* 20% >3-year survival *Disease control:* 12% demonstrated >95% tumour reduction *Immune response:* increased tumour CD8+ and T-bet+ cells; decreased TIM-3+ cells	[31]
	EnAd	Ad11p/3 chimera generated through directed evolution		Ovarian Phase I	IV	*Survival:* 64% PFS *Disease control:* 10% ORR; 35% achieved stable disease; 65% saw reduction in tumour burden; *Immune responses:* 83.3% demonstrated increased CD8+ TILs	[42]
	LoAd-703	Ad5 with 24 bp *E1a* deletion; Pseudo-typed Ad35 knob	TMZ-CD40L; 4-1BBL	PDAC Phase I/II	IT	*Survival:* OS 8.7 months *Disease control:* 44% ORR; 94% DCR *Immune response:* increased effector memory T cells; decreased Tregs and MDSCs	[40]
	ONCOS-102	Ad5 with 24 bp *E1a* deletion; Pseudo-typed Ad3 knob	GM-CSF	Solid tumours Phase I	IT	*Immune response:* increase in TILs; increase in systemic tumour-specific CD8+ T cells; increased tumour PD-L1 expression	[33]
	Telomelysin	Ad5 with *E1a* under hTERT promoter		Oesophageal Phase I	IT	*Disease control:* 91.7% ORR; 83.3% Stage I and 60% Stage II/III CRR *Immune response:* increased tumour CD8+ T cells; increased tumour PD-L1 expression	[39]
	VCN-01	Ad5 with 24 bp *E1a* deletion; *E2F1* promoter insertion; RGDK integrin-binding motif	Hyaluronidase	PDAC Phase I	IT	*Disease control:* injected tumours reduced in size or remained stable; reduction in tumour stiffness	[43]
					IV	*Disease control:* 40–45% ORR including 1 complete response *Immune response:* CD8+ T cell tumour infiltration and IDO upregulation in 64% of patients	[48]
Coxsackie virus	CVA21			NMIBC Phase I	IVS	*Disease control:* 1/15 demonstrated CR; viral protein detected in 86% of tumours with no viral protein seen in stroma *Immune response:* CR patient demonstrated increased immune infiltration; RNA-seq demonstrated increased intrinsic apoptotic cell death pathway and PD-L1, LAG-3 and IDO within the TME	[41]
Herpes simplex virus	T-VEC	HSV1 with *ICP34.5* deletion; *US11* deletion	GM-CSF	Melanoma Phase III	IT	*Survival:* median OS 23.3 months *Disease control:* 19% DRR; 31.5% ORR; 50% demonstrated CR of which 88.5% were estimated to survive at 5-years; median time to CR 8.6 months Approved for the local treatment of unresectable metastatic stage IIIB/C–IVM1a melanoma in Europe and US	[26]
	G207	HSV1 with *ICP34.5* deletion; *UL39* deletion;		GBM Phase I (+Rad)	IT	*Survival:* median OS 7.5 months *Disease control:* median PFS 2.5 months; 67% demonstrated stable or partial response at ≥ 1 time point	[37]
				Paediatric glioma Phase I	IT	*Survival:* Median OS 12.2 months; 36% still alive at 18 months *Disease control:* 18% demonstrated stable disease at 12 months *Immune response:* increased CD4+ and CD8+ T cell tumour infiltration	[44]
	G47Δ	G207 with additional *α47* deletion; *US11* promoter deletion		GBM Phase II	IT	*Survival:* median OS 20.2 months; 84.2% survival at 12 months *Disease control:* median PFS 4.7 months; stable disease in 18 patients at 2 years *Immune response:* increased CD4+, CD8+ and decreased Foxp3+ TIL Conditional and time-limited approval for treatment of GBM in Japan	[27]
	HF10	HSV1 with *UL43*, *UL49.5*, *UL55* & *UL56* deletions; Latency-associated transcripts deletions; *UL53* & *UL54* overexpression		Pancreatic cancer Phase I	IT	*Survival:* median OS 15.5 months; 2 patients were alive at 3 year follow up *Disease control:* median PFS 6.3 months; 33.3% PR; 44.4% SD; 2 patients demonstrated surgical CR *Immune response:* increased CD4+, CD8+ TILs	[45]
				Superficial solid tumours Phase II	IT	*Disease control:* 33.3% SD; 1 patient demonstrated pathological CR after 4 months; 30–61% reduction in tumour size in those demonstrating responses	[36]
	Seprehvir	HSV1 with *ICP34.5* deletion		Paediatric solid tumours Phase I	IT	*Survival:* median OS 7 months *Disease control:* 80% demonstrated SD at 14 days; 43% SD at 28 days	[30]
	OrienX010	HSV1 with *ICP34.5* deletion; *US12* deletion	GM-CSF	Melanoma Phase I	IT	*Survival:* median OS 19.2 months *Disease control:* median PFS 2.9 months; 54.6% of injected tumours regressed, 25.8% of which regressed by ≥30%; 54.1% of non-injected regional tumours regressed, 32.8% of which regressed by ≥30%; 1 distant non-injected metastases regressed by 58%	[38]
	OH2	HSV2 with *ICP34.5* & *ICP47* deletion;	GM-CSF	Solid tumours Phase I/II	IT	*Disease control:* 1 PR; 33% stable disease *Immune response:* 79% saw increased CD8+ TILs; 86% increased CD3+ TILs; 71.4% increased PD-L1+ cells	[46]
Newcastle disease virus	PV701			Solid tumours Phase I	IV	*Disease control:* 61% PFS at 4 months; 33% OR; 1 CR cervical cancer; 2 PRs colorectal; 1 PR melanoma	[8]
Measles virus	MV-CEA		Carcinoembryonic antigen	Ovarian cancer Phase I/II	IP	*Survival:* median OS 12.15 months *Disease control:* 67% SD; 36% demonstrated >30% tumour reduction	[9]
				GBM Phase I	IT	*Survival:* median OS 11.6 months *Disease control:* 59% 3-month PFS; 23% 6-month PFS	[47]
	MV-NIS		Sodium iodide symporter	Ovarian cancer Phase I	IP	*Survival:* median OS 26.2 months *Disease control:* 81% SD	[32]
Vaccinia virus	GL-ONC1		Β-galactosidase; β-glucuronidase	Ovarian cancer Phase I	IP	*Disease control:* median PFS 11.6 months; 78% 6-month PFS; 63% ORR; 52% CR *Immune response:* increased CD4+ & CD8+ TILs	[29]
	JX-594	*TK1* deletion	GM-CSF	HCC Phase II	IT	*Survival:* median OS 9 months; ~35% alive at 2 years *Disease control:* 46% demonstrated tumour control at 8 weeks; average 32.2% decrease in tumour size *Immune response:* increased tumour specific CD8+ TILs	[28]
				HCC Refractory to Sorafenib treatment Phase IIb	IT	*Survival:* no significant increase in survival compared to BSC *Disease control:* 13% DCR compared to 18% DCR with BSC *Immune response:* OV treated patients demonstrated a significant increase in vaccinia-specific T cells; 21.7% OV treated patients demonstrated tumour associated antigen-specific T cells	[50]
Vesicular stomatitis virus	VSV- IFNβ-NIS		IFN-β; sodium iodide symporter	TCL Phase I	IV	*Disease control:* 1 6-month PR; 1 20-month CR; 71.4% reduction in ≥1 tumour	[35]

Abbreviations: Ad, adenovirus; BSC, best standard care; CR, complete response; CRR, complete response rate; DCR, disease control rate; DRR, disease response rate; GBM, glioblastoma; GM-CSF, granulocyte-macrophage colony-stimulating factor 2; HCC, hepatocellular carcinoma; HSV, herpes simplex virus; IDO, indoleamine 2,3-dioxygenase; IP, intraperitoneal; IT, intra-tumoural; IV, intravenous; IVS, intravesicular;LAG-3, lymphocyte activation gene 3; MDSC, myeloid-derived suppressor cell; NDV, Newcastle disease virus; NMIBC, non-muscle-invasive bladder cancer; MV, measles virus; ORR, overall response rate; OS, overall survival; PDAC, pancreatic ductal adenocarcinoma; PD-L1, programmed death-ligand 1; PFS, progression free survival; PR, partial response; SD, stable disease; TCL, T cell lymphoma; TIL, tumour-infiltrating lymphocyte; TIM-3, T-cell immunoglobulin and mucin domain 3; Treg, regulatory T cell; TMZ-CD40L, trimerized membrane-bound extracellular CD40L VSV, vesicular stomatitis virus; VV, vaccinia virus.

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
