# Peer review of "Oncolytic Viruses and Immune Checkpoint Inhibitors: The “Hot” New Power Couple"

_cancers, 2023, doi:10.3390/cancers15164178_

Round 1
Reviewer 1 Report
The manuscript by Lovatt and Parker reviews the recent literature regarding the therapeutic relationship between oncolytic viruses and immune checkpoint inhibitors and their combination for cancer therapy. Overall, this is a really well written review article with a lot of helpful information for people working in the field of oncolytic virotherapy. The text is detailed, well balanced and the exhaustive work done by both the authors will ensure that this review article will be read and cited by many.
I list below a few minor recommendations that may improve the manuscript before its publication.
1) Lines 110-111: The antiviral response and its frequent defects in cancer (e.g. the type I interferon pathway) may be cited here as it is a major determinant for the sensitivity of malignant cells to oncolytic viruses (especially RNA viruses).
2) In table 1, considering the particular shape of VSV particles, I would indicate their diameter to be 70-200nm.
3) Lines 53-58 and 132-139: The authors quickly explain the differences between 'cold' and 'hot' TMEs and how OVs can 'heat' them. The reality is a little more complex than this cold/hot dichotomy, with other (sub)categories that are commonly described (see PMIDs 29686425, 31974169 for example). A 'cold' TME may thus be an immune desert, be immune-excluded (with T cells kept at the tumour margins), or be infiltrated but with a strong immunosuppression (e.g. MDCSs, Tregs...). 'Hot' tumour are usually described as 'inflamed' (here the categories can also become more complex with the presence or not of TLS...). I believe that quickly explaining this diversity may help the readers to better understand what is at stake with the use of OVS.
4) Line 193 and Table 2, when citing the G47delta virus, please mention that the clinical approval in Japan for glioblastoma treatment is still (to my knowledge) conditional and time-limited (pending results of more advanced clinical trials). There is also a typo (‘approvd’) in the table.
5) Tables 2, 3 and 4 will be a precious source of information for future readers. However, I feel that the important results are sometimes a lost in the impressive amount of information collected by the authors. Could they simplify the presentation of the key findings? Maybe using predefined columns (survival data for OS/PFS…, disease evolution for CR/PR…, immune consequences…) to better organize it? Switching to a landscape display (if authorized by the editor) could help to fit everything in.
6) In Table 2, the authors could mention that JX-594 failed in phase III trial in combination with sorafenib (PMID 31413923). Also, the abbreviation for TMZ is lacking.
7) At the end of part 4.2, the authors may add a few lines on the potential drawbacks of using ICIs encoded in OVs. For instance, whether the location of OV replication is really the best to produce ICIs, as some data show that these may act elsewhere too? How can you stop ICIs being produced if toxicities are observed? Does this add to the complexity for producing/characterizing the recombinant OVs? What about approval by regulatory agencies?
Author Response
The manuscript by Lovatt and Parker reviews the recent literature regarding the therapeutic relationship between oncolytic viruses and immune checkpoint inhibitors and their combination for cancer therapy. Overall, this is a really well written review article with a lot of helpful information for people working in the field of oncolytic virotherapy. The text is detailed, well balanced and the exhaustive work done by both the authors will ensure that this review article will be read and cited by many.
I list below a few minor recommendations that may improve the manuscript before its publication.
Author response: We thank the reviewer for their kind comments regarding our manuscript, for describing it as “really well written with a lot of helpful information”, and describing the work as “detailed” “well balanced” and “exhaustive”. We agree that this article will be of significant interest to those in the field and will become well read and cited. We additionally thank the reviewer for their helpful comments which we have taken on board and addressed, which have improved the article.
1) Lines 110-111: The antiviral response and its frequent defects in cancer (e.g. the type I interferon pathway) may be cited here as it is a major determinant for the sensitivity of malignant cells to oncolytic viruses (especially RNA viruses).
Author response: This is an important point, and we have revised the manuscript - lines 115-117 now address that defects in anti-viral responses in tumour cells give some OVs a replicative advantage.
2) In table 1, considering the particular shape of VSV particles, I would indicate their diameter to be 70-200nm.
Author response: We amended that table as suggested.
3) Lines 53-58 and 132-139: The authors quickly explain the differences between 'cold' and 'hot' TMEs and how OVs can 'heat' them. The reality is a little more complex than this cold/hot dichotomy, with other (sub)categories that are commonly described (see PMIDs 29686425, 31974169 for example). A 'cold' TME may thus be an immune desert, be immune-excluded (with T cells kept at the tumour margins), or be infiltrated but with a strong immunosuppression (e.g. MDCSs, Tregs...). 'Hot' tumour are usually described as 'inflamed' (here the categories can also become more complex with the presence or not of TLS...). I believe that quickly explaining this diversity may help the readers to better understand what is at stake with the use of OVS.
Author response: We agree with this suggestion and thank the reviewer for this point. As suggested, we added a section on the subcategories of “cold” TMEs (lines 53-62). In addition, we also added a line at 137 about increasing T cell infiltration into “cold” TMEs.
4) Line 193 and Table 2, when citing the G47delta virus, please mention that the clinical approval in Japan for glioblastoma treatment is still (to my knowledge) conditional and time-limited (pending results of more advanced clinical trials). There is also a typo (‘approvd’) in the table.
Author response: We have amended this as suggested (see line 195-197 and in Table 2). We thank the reviewer for noting the typo, which has also been corrected.
5) Tables 2, 3 and 4 will be a precious source of information for future readers. However, I feel that the important results are sometimes a lost in the impressive amount of information collected by the authors. Could they simplify the presentation of the key findings? Maybe using predefined columns (survival data for OS/PFS…, disease evolution for CR/PR…, immune consequences…) to better organize it? Switching to a landscape display (if authorized by the editor) could help to fit everything in.
Author response: We thank the reviewer for highlighting the value in these tables and for the suggestions to improve their impact further. These are good suggestions we have taken on board in the revised manuscript. We have added “survival”, “disease control” and “immune response” subheadings within the key findings column to help discern the findings of the studies. We also present the tables in landscape format as suggested.
6) In Table 2, the authors could mention that JX-594 failed in phase III trial in combination with sorafenib (PMID 31413923). Also, the abbreviation for TMZ is lacking.
Author response: We added this trial to the table, along with a comment on the reduced efficacy of OV monotherapy in treatment-refractory patients on lines 207-211. We have added the abbreviation for TMZ to the footnote.
7) At the end of part 4.2, the authors may add a few lines on the potential drawbacks of using ICIs encoded in OVs. For instance, whether the location of OV replication is really the best to produce ICIs, as some data show that these may act elsewhere too? How can you stop ICIs being produced if toxicities are observed? Does this add to the complexity for producing/characterizing the recombinant OVs? What about approval by regulatory agencies?
Author response: Again, this is a helpful suggestion. We have added a section on the potential limitations of ICI-OVs (lines 407-422). To address the reviewer’s other specific queries (which did not appear to warrant :
- the addition of an ICI as a transgene uses the same techniques as the insertion of any transgene into a recombinant OV and does not add any additional complexity to the production or characterisation of the OV.
- From first-hand experience of working with regulators to take an OV encoding an ICI to clinic, we have noted that the use of transgenes encoding anticancer drugs with known/acceptable toxicity profiles is in fact looked upon favourably by the regulatory agencies, since the transgene product is known and has an approved profile. Therefore, these drugs have already gone through extensive testing in regard to their safety profiles. Our “in house” calculations have indicated that the total amount of ICI expressed if every tumour cell were transduced simultaneously indicate that this would still produce less ICI than then recommended IV dose. We have resisted the temptation here to add a section related to our own work here which is developing OV that can only infect transformed cells not health cells (see for e.g. https://pubmed.ncbi.nlm.nih.gov/29798908/, https://pubmed.ncbi.nlm.nih.gov/30479689/ and https://pubmed.ncbi.nlm.nih.gov/34066836/) but we do agree that the development of technologies with improved “on target” and reduced “off target” activities will be critical to this strategy. However, we have written about this in previous reviews and so have opted not to discuss this at length here.
Reviewer 2 Report
The manuscript mainly described the application status and some practice issues for OVs in cancer treatment and few to discuss the role of ICI. They also aim to highlight the combination therapy of ICI and OVs in cancer treatment. The manuscript provided interesting informations for the research field of OVs and ICI. However, the authors need some modification for current manuscript before it was accepted for publication in CANCERS.
1) The structure of the manuscript need modification. They mainly addressed the OVs but few on ICI, and obviously the manuscript have two parts: OVs and combination therapy. Therefore, I suggest the authors should re-organize the structure:
2. OVs:
2.1 Oncolytic viruses
2.2 Oncolytic virus monotherapy
2.3 Turning cold tumours hot: the OV immune response
3. Combined OV and ICI therapy
2) The author should add more information for the current status of Oncolytic virus monotherapy especially the disadvantage of monotherapy which could be solve by the combination with ICI therapy.
3) The section of Combined OV and ICI therapy is an important part for the manuscript. The author should modify or discuss the ICI therapy and some problem for ICI therapy which could be solved by the combination of OV and ICI therapy.
4) Combined OV and ICI therapy: this section seems not well organized and the author should pay more attention to the first part: How and what is the status of Combined OV and ICI therapy? They just provided a short paragraph for this.
5) Combined OV and ICI therapy: the author need to re-organize this section by adjust the sub-sections and make it more easily to read and understand.
6) The author need provide an abbreviate list if they do not provide the whole name at the first place the abbreviate appears.
Author Response
The manuscript mainly described the application status and some practice issues for OVs in cancer treatment and few to discuss the role of ICI. They also aim to highlight the combination therapy of ICI and OVs in cancer treatment. The manuscript provided interesting informations for the research field of OVs and ICI. However, the authors need some modification for current manuscript before it was accepted for publication in CANCERS.
Author response: We thank the reviewer for their time and effort in reviewing our manuscript and for highlighting the “interesting information” that our review providing. We have carefully considered their suggested alterations and respond to their suggestions below.
1) The structure of the manuscript need modification. They mainly addressed the OVs but few on ICI, and obviously the manuscript have two parts: OVs and combination therapy. Therefore, I suggest the authors should re-organize the structure:2. OVs, 2.1 Oncolytic viruses, 2.2 Oncolytic virus monotherapy, 2.3 Turning cold tumours hot: the OV immune response, 3. Combined OV and ICI therapy
Author response: We thank the reviewer for their suggestion to restructure the manuscript but feel that the structure as it stands works the best.
We believe that the “turning cold tumours hot: the OV immune response” section contains all the relevant information that is necessary for the reader to understand how OVs work as therapies and describes the immune responses they induce in detail, thus allowing the reader to better understand the findings of these OV monotherapy trials. This OV immune response section discusses how the OV therapies increase the immune response within the TME and the mechanisms and immune cells involved in this response. We believe that this information is important for the reader to understand and appreciate what the changes in the immune responses observed in the OV monotherapy trials mean in terms of patient benefit.
We are also conscious that reviewer 1 commended the structure of the manuscript (and awarded a mark of 5/5 stars for the section of being “well organised”) and on balance we agree with reviewer 1. Any significant alteration to the structure may detrimentally impact the scores as provided by this reviewer.
2) The author should add more information for the current status of Oncolytic virus monotherapy especially the disadvantage of monotherapy which could be solve by the combination with ICI therapy.
Author response: We thank the reviewer for their suggestion and highlight that these are described at length in the following sections:
- Lines 191-211 and table 2 discuss the current status of OV monotherapies and covers the virus vectors used, the OVs which have been given approval and the results of 26 clinical trials on OV monotherapies. Lines 213-221 discuss the limitations of OV monotherapies and the mechanisms which may contribute to the modest effects seen in most of these trials. The purpose of using OVs in combination with ICIs is less to increase the efficacy and clinical responses to OV monotherapies, but more to use OVs as a mechanism to “prime” the TMEs of patients who would not normally respond to ICI therapies, this is discussed further in lines 63-75.
- We have additionally added a study (reference 51) in table 2 and lines 207-2011 in which OV monotherapy failed to show ay clinical responses, thus highlighting that OV monotherapies may not work in patients with treatment-refractory disease.
3) The section of Combined OV and ICI therapy is an important part for the manuscript. The author should modify or discuss the ICI therapy and some problem for ICI therapy which could be solved by the combination of OV and ICI therapy.
Author response: Lines 40-52 and 240-265 discuss the current status of ICI therapies, whilst lines 52-62 and 265-272 discuss the current limitations of ICI therapies, mainly that a large proportion of patients treated with these therapies do not respond due to a lack of a pre-existing immune response and a “cold” TME lacking in TILs. Lines 63-68, lines 272-281 and figure 2 discuss how OVs can help to overcome these limitations by “heating up” the TME by triggering an “in situ” tumour vaccination through the release of tumour antigens in the presence of virus-induced inflammation, thus sensitising tumours in non-responder patients to ICI therapy. We therefore consider that this is amply covered in the manuscript.
4) Combined OV and ICI therapy: this section seems not well organized and the author should pay more attention to the first part: How and what is the status of Combined OV and ICI therapy? They just provided a short paragraph for this.
Author response: We are unsure which paragraph in section 4.1 the reviewer is referring to. Section 4.1 and table 3 cover the current status of adjuvant OV and ICI trials and summarise the key findings of 11 trials as well as the markers of response identified in these trials (4.1.1) and the adverse events (4.1.2) observed. Section 4.2 and table 4 explore the current status of research into OVs encoding ICIs and summarise the findings of 24 pre-clinical trials in addition to additional ICI targets which could be explored in the future (4.2.1).
To address and expand on the reviewer’s point, we added a section in lines 404-419 to discuss the potential limitations of creating OVs which encode for ICIs to give a more complete review of the current status of combined OV and ICI therapies and the challenges that will need to be addressed in future research. We hope this addition adequately addresses the reviewer’s point.
5) Combined OV and ICI therapy: the author need to re-organize this section by adjust the sub-sections and make it more easily to read and understand.
Author response: Section 4 is comprised of: 4.1 neoadjuvant therapies, 4.1.1 markers of response, 4.1.2 adverse events, table 3 neoadjuvant OV and ICI trials and their key findings, 4.2 OVs encoding ICIs, 4.2.1 additional targets and table 4 pre-clinical trials of OVs encoding ICIs and their key findings. We feel that the current structure of this section works well and that re-structuring the subsections would negatively impact the readability of this section. Furthermore, reviewer 1 commended us on the structure of the paper and we fear that reorganising the paper may negatively impact their review.
6) The author need provide an abbreviate list if they do not provide the whole name at the first place the abbreviate appears.
Author response: We thank the reviewer for pointing this out. We have carefully checked the manuscript and ensured that every abbreviation is expanded at the first instance it appears in the manuscript. The addition of a specific abbreviation list is not allowable or recommended by the journal.
Round 2
Reviewer 2 Report
The authors addressed most of my concerns and I do not have further comments.